# Clinical Importance of Drug Adherence during Tyrosine Kinase Inhibitor Therapy for Chronic Myelogenous Leukemia in Chronic Phase

**Yasuhiro Maeda \*** , **Atsushi Okamoto, Kenta Yamamoto, Go Eguchi and Yoshitaka Kanai**

Department of Hematology, National Hospital Organization Osaka Minami Medical Center, Osaka 586-8521, Japan; atsokmt@ommc-hp.jp (A.O.); yamaken@ommc-hp.jp (K.Y.); g.eguchi@ommc-hp.jp (G.E.); y_kanai@ommc-hp.jp (Y.K.)
**\*** Correspondence: ymaeda@ommc-hp.jp; Tel.: +80-(0)7-2153-5761 (ext. 4111)

**Abstract:** Chronic myeloid leukemia (CML) is a myeloproliferative neoplasm constituting approximately 15% of newly diagnosed leukemia in adult patients. Development of tyrosine kinase inhibitors (TKIs) have dramatically improved outcomes in patients with chronic CML in chronic phase. However, adverse drug events (ADEs) associated with TKI therapy have influenced drug adherence, resulting in adverse clinical outcomes and a decline in the quality of life (QoL). In this study, we carried out a unique questionnaire survey to evaluate ADEs, which comprised 14 adverse events. We compared drug adherence rates between patients using imatinib and those who switched from imatinib to nilotinib, a second-generation TKI. Following the switch, the total number of ADEs decreased considerably in most cases. Simultaneously, better QoL was observed in the nilotinib group than in the imatinib group. Drug adherence was measured using Morisky's 9-item Medication Adherence Scale (MMAS). MMAS increased significantly after switching to nilotinib in all cases. Drug adherence is a critical factor for achieving molecular response in patients with CML. In fact, our results showed a strong inverse correlation between clinical outcome (international scale (IS)) and adherence (MMAS), with a stronger tendency in the nilotinib group than in the imatinib group. In conclusion, low occurrence of ADEs induced a high level of QoL and a good clinical response with second-generation TKI nilotinib treatment.

**Keywords:** chronic myeloid leukemia; tyrosine kinase inhibitor; adverse drug event; quality of life; second-generation tyrosine kinase inhibitor; nilotinib

---

## 1. Introduction

Chronic myeloid leukemia (CML) is characterized by the translocation between the 9th chromosome and the 22nd chromosome. The treatment of CML has dramatically changed by tyrosine kinase inhibitors (TKIs). Furthermore, outcomes in patients with CML in chronic phase has been improved [1]. Several adverse drug events (ADEs) related to TKI in CML patients are common, including general edema, nausea, fatigue, and musculoskeletal symptoms. It has been reported that ADEs had a negative influence on their daily quality of life (QoL). However, new ADEs developed once imatinib (IM) was switched to nilotinib (NILO); therefore, early and successful management of ADEs is required for the acquisition of tolerance to treatment [2]. The ADEs (peripheral edema, muscle spasm, and eruption) that occurred at the beginning of IM treatment disappeared after switching to NILO [3].

Adherence is compliance with a medication regimen and is based on patient understanding, decision-making, and therapeutic cooperation. Adherence is defined by various factors. It has been

reported that drug adherence has been reported as a critical factor for achieving molecular response in patients with CML [4–7], and non-adherence to TKI therapy may influence the disease outcome [5]. In this study, we evaluated drug adherence using the 9-item Morisky Medication Adherence Scale (MMAS) [8,9]. We compared drug adherence rates between patients using IM and those who switched from IM to NILO, a second-generation TKI. Drug adherence determined with 9-item MMAS was improved significantly in the NILO group compared IM group. Guérin et al. [10] reported that among the patients treated with second-line TKIs, those treated with NILO had a significantly higher adherence compared to patients treated with dasatinib. However, Trivedi et al. reported that among the second-line TKI-treated CML patients, dasatinib patients had significantly higher adherence and lower discontinuation rates compared with those receiving second-line nilotinib [11]. Chen et al., however, raised some questions about the results described above [12]. In general, social, disease-related, treatment-related, and patient-centered factors contribute to improved adherence [5,13]. However, no significant differences in adherence, hospitalization, or emergency room visits have been reported among patients initiating a second vs first-generation TKI [14]. Marin et al. reported that no complete molecular responses were observed when adherence was ≤90%, and no major molecular responses were observed when adherence was ≤80%; the adherence rate for each patient was defined as the dose taken according to the microelectronic monitoring systems (MEMS) reading and expressed as a percentage of the dose prescribed during the total duration of the study [5]. We, therefore, examined the relationship between clinical outcome using the international scale (IS) for *BCR/ABL* and clinical adherence determined by MMAS. This study aimed to evaluate patient-reported ADEs during TKI treatment and their influence on adherence and QoL in CML patients in chronic phase.

## 2. Patients and Methods

### 2.1. Patient Population

Twenty patients with CML, who received TKIs for at least 6 months at the National Hospital Organization Osaka Minami Medical Center, were selected for this study. Twelve patients were male and median age of all patients was 58 years (range: 28–80 years). Socal score was low and intermediate in 18 (90%) and 2 (10%) patients, respectively. We compared drug adherence rates and QOL between patients using imatinib and those who switched from imatinib to nilotinib, a second-generation TKI. All study participants provided informed consent, and the study design was approved by the appropriate ethics review board.

### 2.2. Adverse Events

All patients were questioned using an interview form (Figure 1). Patient-reported ADEs were assessed during the interview using a structured questionnaire (Figure 1). Fourteen ADEs were included in the form.

## \<Questionnaire for changing AE\>

★Please fill in check mark in the place to apply to it.

| | Severity score | | | | | | | |
| | During imatinib treatment | | | | Present | | | |
| | No AE | Low | Intermediate | High | No AE | Low | Intermediate | High |
|---|---|---|---|---|---|---|---|---|
| Face Edema | | | | | | | | |
| Peripheral Edema | | | | | | | | |
| Lids Edema | | | | | | | | |
| Chest Discomfort | | | | | | | | |
| Headache | | | | | | | | |
| Genaral Fatigue | | | | | | | | |
| Depression | | | | | | | | |
| Diarrhea | | | | | | | | |
| Constipation | | | | | | | | |
| Nausea | | | | | | | | |
| Muscle Pain | | | | | | | | |
| Muscle Cramp | | | | | | | | |
| Eruption | | | | | | | | |
| Pruritus | | | | | | | | |
| Oher（　　） | | | | | | | | |

**Figure 1.** An interview form for adverse drug events (ADEs) was prepared for all patients. Patient-reported adverse events (ADEs) were assessed during the interview using a structured questionnaire.

### 2.3. Quality of Life (QoL)

QoL was judged on a scale of 1 to 5 during IM treatment in the past and during NILO treatment in the present (Figure 2).

## \<Questionnaire for QOL\>

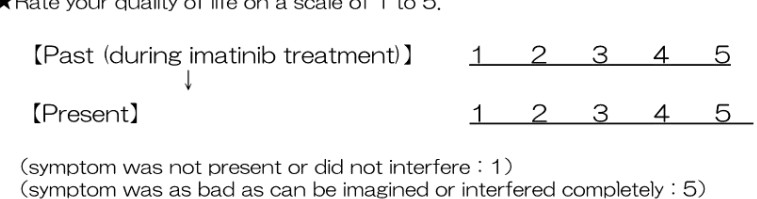

**Figure 2.** An interview form for quality of life (QoL) was prepared for all patients. QoL was judged on a scale 1 to 5 during imatinib (IM) treatment (in the past) and during nilotinib (NILO) treatment (in the present).

### 2.4. Assessment of Adherence and Clinical Outcome

Patient adherence was measured using the 9-item Morisky Medication Adherence Scale (MMAS) [8,9], with scores ranging from 1–13, where 13 indicates perfect adherence. MMAS is composed of nine questions that explore the adherence behavior based on forgetfulness, negligence, interruptions in drug intake, and the restart of drug intake. Patients with an MMAS score of 11 or above were classified as adherent [15].

### 2.5. Statistical Analysis

Statistical analysis was performed using GraphPad Prism v6 software (GraphPad software Inc., La Jolla, CA, USA). Unpaired student *t* test (Mann-Whitney test) was used comparison between two groups. *p* value of ≤0.5 was considered significant.

### 2.6. Ethics Committee Approval and Patient Consent

All study participants provided informed consent. The study design was approved by the appropriate ethics review board.

## 3. Results

### 3.1. Comparison of ADEs between IM and NILO Treatments

A questionnaire survey (Figure 1), which included 14 adverse events, was carefully carried out by calculating system. After switching from IM to NILO, the total number of AEs decreased in most cases, except in 2 (Figure 3). New AEs developed upon switching to NILO; however, tolerance was gradually acquired by management of AE.

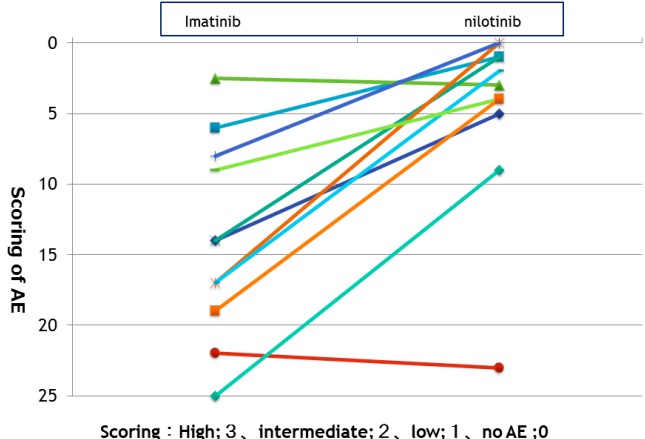

**Figure 3.** A unique questionnaire survey including 14 ADEs was carefully conducted by calculating the severity score. Comparison of the ADE scores between IM group and NILO group.

### 3.2. Improved Symptoms after Switching from Imatinib to Nilotinib

We investigated the type of ADEs that improved upon switching to NILO. As shown in Figure 4, ADEs such as facial edema, peripheral edema, lids edema, general fatigue, depression, nausea, muscle pain, and muscle cramp, were reduced significantly. These results indicated that fluid retention, digestive symptom, and muscle symptom induced by IM improved upon switching to NILO.

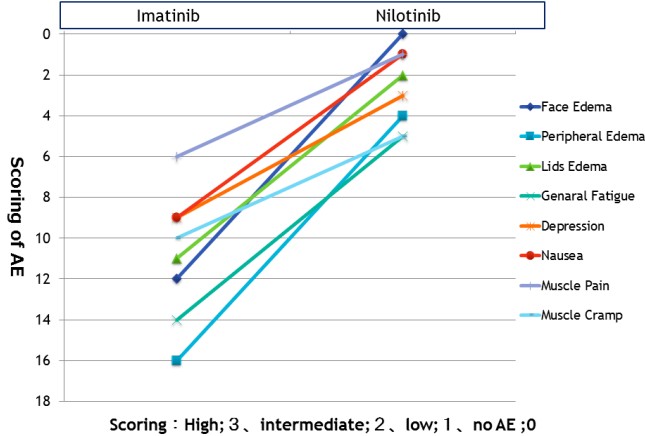

**Figure 4.** The ADEs which improved by switching to NILO are indicated. ADEs described below including facial edema, peripheral edema, lids edema, general fatigue, depression, nausea, muscle pain, and muscle cramp improved.

### 3.3. Alteration in QoL upon Switching from Imatinib to Second-Generation TKI

As the ADEs induced by IM was reduced by NILO administration, change of QoL was examined by a questionnaire study (Figure 2). The QoL score was significantly decreased, indicating an improved QoL upon NILO administration (Figure 5).

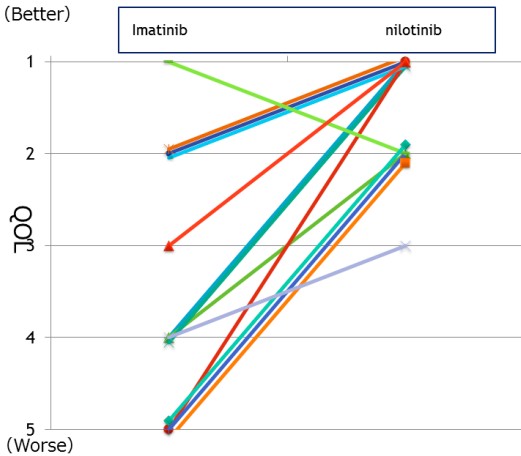

**Figure 5.** An interview form for QoL was prepared for all patients. Comparison of QoL score between IM group and NILO group.

*3.4. Relationship between Clinical Outcome (IS) and Drug Adherence (MMAS) in the NILO Group Compared to That in the IM Group*

We compared the drug adherence rates and clinical outcome. Drug adherence was measured using MMAS and clinical outcome was evaluated based on IS. A significant relationship between clinical outcome and drug adherence was found in the NILO group ($p = 0.0002$) (Figure 6A) than in the IM group ($p = 0.0024$) (Figure 6B).

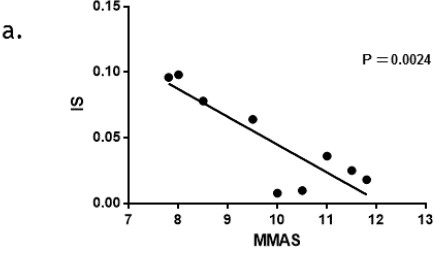

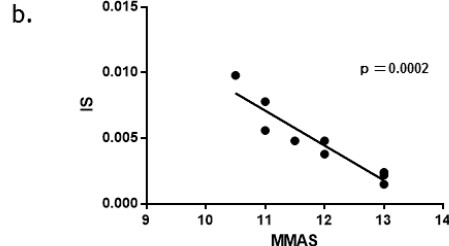

**Figure 6.** Relationship between drug adherence rates and clinical outcome determined with IS. Drug adherence was measured by Morisky's 9-item Medication Adherence Scale (MMAS). A more significant relationship between those evaluated parameters was found in the NILO group ($p = 0.0002$) (Figure 6a) than in the IM group ($p = 0.0024$) (Figure 6a). The straight line has a steeper slope in NILO group than IM group. Figure 6 divided into a and b. As shown in Figure 6 legend, Figure 6a showed relationship between drug adherence rates and clinical outcome determined with IS in IM group, and Figure 6b showed that in NILO group.

## 4. Discussion

Several ADEs related to TKI in CML during chronic phase, including general edema, nausea, fatigue, and musculoskeletal symptoms, occur at varying frequencies depending on the TKIs. We

created a specific questionnaire survey that included 14 ADEs and was carefully conducted by calculating severity scores. After switching from IM to NILO, the total number of ADEs decreased in most cases. Although new ADEs developed initially after switching to NILO, tolerance was acquired by management of ADEs [3,16]. In particular, fluid retention, digestive symptoms, and muscle symptoms including facial edema, peripheral edema, lids edema, general fatigue, depression, nausea, muscle pain and muscle cramps induced by IM were reduced significantly (Figure 4). Kekäle et al. reported that they were unable to find a clinical correlation between these symptoms and patient adherence although, they did find a significant correlation between higher number of symptoms and a negative impact on the patient's QoL [17]. Furthermore, they reported that intentional non-adherence was more common in women than in men (37% and 24%) and in patients receiving dasatinib and NILO than in patients receiving IM (44%, 44% vs 26%, respectively) [17]. However, Rychter et al. reported that there were no differences in adherence among patients treated with imatinib, dasatinib, and nilotinib ($p = 0.249$) [18]. In our study, the QoL score was significantly decreased in most patients who switched to NILO, which might be the result of fewer ADEs. Previously, we reported that statistically significant differences in adherence, defined by an MMAS score of ($p = 0.0011$), were observed between the IM and NILO groups [19]. It has been reported that adherence is the most critical factor for achieving clinical response and ultimately for improving survival in patients with CML receiving TKI therapy [4,14]. Winn et al. reported that, in a multivariate analysis, individuals with cost-sharing subsidies, younger age, lower comorbidity, and later year of diagnosis were significantly more likely to initiate TKIs [20]. We also compared drug adherence rates and clinical outcomes. Clinical outcomes were evaluated using the IS for major *BCR/ABL* gene expression. Significant relationships between clinical outcome and drug adherence rates were found in the IM ($p = 0.0024$) and NILO ($p = 0.0002$) groups, with a more significant tendency in the NILO group. These results might be due to the difference in drug adherence between the TKI groups. It has been reported that the Morisky high adherence was positively associated with complete hematologic remission in the chronic phase of CML [16,21]. Drug adherence has been reported as a critical factor for achieving molecular response in patients with CML [4–7], and non-adherence to TKI therapy may influence the disease outcome [5].

## 5. Conclusions

Various factors have been assessed for their impact on drug adherence. Among the factors, ADEs of TKI have significant influence on drug adherence results, leading to poorer outcomes during the clinical course and a decline in the QoL. Management of ADEs associated with TKI treatment is the most important strategy to maintain a high-drug adherence. Furthermore, drug adherence has been reported to be a critical factor for achieving molecular response in patients with CML. In fact, our results showed a strong inverse correlation between clinical outcome and adherence.

**Author Contributions:** conceptualization, Y.M.; methodology, Y.M.; formal analysis, A.O.; investigation, K.Y.; resources, G.E.; data curation, Y.M.; writing—original draft preparation, Y.M.; writing—review and editing, Y.M.

**Funding:** This research received no external funding.

**Acknowledgments:** The authors thank H. Sakamoto for technical support and preparing the manuscript.

**Conflicts of Interest:** The authors declare no conflict of interest.

## Abbreviations

| | |
|---|---|
| CML | Chronic myeloid leukemia |
| TKI | Tyrosine kinase inhibitors |
| ADE | Adverse drug events |
| QoL | Quality of life |
| MMAS | Morisky Medication Adherence Scale |
| IM | Imatinib |
| NILO | Nilotinib |

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
