# Peer review of "Clinical Importance of Drug Adherence during Tyrosine Kinase Inhibitor Therapy for Chronic Myelogenous Leukemia in Chronic Phase"

_reports, doi:10.3390/reports2040025_

Round 1
Reviewer 1 Report
The authors present an interesting Case Report about a tricky problem. They focused their attention and analysis on a specific disease , Chronic Myelogenous Leukemia (CML) today cured with a single drug and looking at their quality of life that could condition the final successful result. I don't have criticism about the small sample of recruited patients (the type of disease is quite rare) and the good methodology followed by Authors but before making elegible for consideration the paper I would like to better understand and have back a reply on the following points: 1. Adherence is the most critical factor for achieving clinical response and ultimately for improving survival in patients with CML receiving TKI therapy. There is a consistent difference using Nilotimib versus Imatimib.Is the hemathological response identical using both drugs? 2. If yes why don't you recommend to use Nilotimib since the beginning instead of Imi? 3. In the Conclusion you should give a practical suggestion because in the reported literature the data are still controversial about the clinical result in using the 2 drugs. 4. Please explain why we continue to indicate the use of Imatimib in fron line therapy (costs? ADEs? sex?....) instead of NilotimibAuthor Response
In hematological response, nilotinib is more excellent than imatinib, as shown in Figure 6. We recently recommend nilotinib as a first treatment for CML in chronic phase. Figure 6 showed that relationship between drug adherence and clinical outcome. A more significant relationship between them was observed in treatment with nilotinib. Recently, we have started nilotinib treatment in chronic phase CML instead of imatinib.Reviewer 2 Report
The study by Maeda et al. presented clinical outcome after switching from IM to NILO.
The sample size is small. Authors should provide patients clinical information and should analyze the efficacy of switching IM to NILO with age. It will improve their manuscript. Authors should check the legend of Figure 6.Author Response
Clinical informations of those patients were added in text. Legend of Figure 6 was improved.Reviewer 3 Report
Clinical importance of drug adherence during tyrosine kinase inhibitor therapy for CML in chronic phase
In this study Maeda et al. compared drug adherence rates between patients using Imatinib and those who switched from Imatinib to second generation TKI , nilotinib. They report that there is low adverse drug events (ADEs) associated with nilotinib treatment which induced a high level of quality of life and a good clinical response in patients. The study is well designed and executed, however there are major shortcomings in the study that need to be addressed-
Comments –
The variables: gender, age, education, place of residence, family circumstances and duration of therapy should be taken into account when assessing treatment adherence. The sample size is very small. Only 20 patients with CML are included in this study. What are the factors responsible for high ADE and low adherence in patients using Imatinib or vice versa in the patients using nilotinib? The molecular response (i.e. amount of BCR-ABL in blood) should be factored in to determine the effect of drug on patients before and after switching to nilotinib.Author Response
1. Nilotinib is improved drug as a second generation TKI. In fact, AE of nilotinib related TKI was decreased significantly in compared with imatinib. As same as molecular response rate is high In treatment with nilotinib as shown in Figure 6.
Round 2
Reviewer 2 Report
The authors have made satisfactory changes in their manuscript.
Reviewer 3 Report
Authors have changed satisfactory changes in the manuscript to improve it. Although, they could not addressed some of the concern like small sample size however, they have addressed other concerns.